# Seeking Causality in the Links between Time Perspectives and Gratitude, Savoring the Moment and Prioritizing Positivity: Initial Empirical Test of Three Conceptual Models

**DOI:** 10.3390/ijerph19084776

**Published:** 2022-04-14

**Authors:** Bozena Burzynska-Tatjewska, Gerald Matthews, Maciej Stolarski

**Affiliations:** 1Faculty of Psychology, SWPS University of Social Sciences and Humanities, 03-815 Warsaw, Poland; bburzynska@st.swps.edu.pl; 2Department of Psychology, George Mason University, Fairfax, VA 22030, USA; gmatthe@gmu.edu; 3Faculty of Psychology, University of Warsaw, 00-183 Warsaw, Poland

**Keywords:** time perspective, gratitude, savoring the moment, prioritizing positivity, well-being

## Abstract

We provide an initial empirical test of three conceptual models reflecting possible patterns of causality effects in the relationships between time perspective (TP), gratitude, savoring the moment, and prioritizing positivity (referred to as well-being boosters, WBBs), and mental well-being. The first one, trait-behavior model, states trait TPs increase the tendency to use specific WBBs in order to increase mental well-being. The second model, the accumulation model, proposes that a regular practice of particular WBBs fosters adaptive TPs which in turn impact well-being. The third model, the feedback loop, suggests that WBBs and positive TPs reciprocally strengthen one another and together contribute to higher mental well-being. Participants (N = 206; M_age_ = 30.90, SD = 8.39, 74% females) filled questionnaires measuring TPs, WBBs, and well-being twice, in a one-year interval. Using cross-lagged panel analyses we examined the direction of causation in the relationships among the variables. Past-Positive had a significant cross-lagged effect on gratitude, Present-Fatalistic had a significant effect on savoring. Both Past-Negative and Present-Fatalistic perspectives displayed significant causal effects on well-being. The results partly support the trait-behavior model. However, given that the second wave was conducted shortly after the onset of the COVID-19 pandemic, further studies are required to better understand the interplay between the studied traits.

## 1. Introduction

Time perspective (TP) is a robust regulatory mechanism associated with numerous indicators of social and emotional adaptation [1,2]. Multiple studies have investigated the associations between TPs and various aspects of well-being [3,4]. However, the mechanisms that are responsible for the existence and robustness of these associations are still little explored. In striving to disentangle these effects, Burzynska and Stolarski [5] emphasized three well-established constructs from the field of positive psychology: gratitude, savoring, and prioritizing positivity, also known as well-being boosters (WBBs). They pointed out that each of these categories has a specific temporal anchoring: gratitude is primarily concerned with the past, savoring is concerned with the present moment, and prioritizing positivity involves prospective preparation. Based on their conceptual analysis they proposed four hypothetical models to explain the interplay between TPs, WBBs, and mental well-being.

One of the models, referred to as the match-mismatch model, postulates an interaction between each of the WBBs and respective positive TP dimension (e.g., gratitude and Past-Positive), such that higher levels of the respective TP should enhance the effects of each of the WBBs on well-being. However, the results of two studies aiming to test the model [6] undermined its major assumption. Both the “positive” TP dimensions and WBBs were associated with higher well-being in bivariate analyses, but the hypothesized interactions between the “positive” TP dimensions and WBBs in predicting well-being were not obtained. Regression models identified multiple TP × WBB interactions but the direction of the effects was typically opposite to expectation from the match-mismatch model. For example, one interaction suggested that the effect of prioritizing positivity on well-being was weaker when future-positive TP was high and stronger for low future-positive individuals. That is, there was no synergistic effect of TPP and WBB on well-being; instead, it seems that prioritizing positivity can compensate for low future-positive TP. Significant interactions involving the “negative” TP dimensions—Past-Negative, Present-Fatalistic, and Future-Negative—also suggested a compensation instead of an enhancement effect (e.g., gratitude attenuated the undesirable effects of Past-Negative). The authors concluded that the remaining conceptualizations may better illustrate the pattern of the associations between TPs, WBBs, and mental well-being.

In the present article, we provide an initial empirical test of the three remaining models proposed by Burzynska and Stolarski [5]: the trait-behavior model, the accumulation model, and the feedback-loop model. To address the key role of causal mechanisms in each of the models, such a test required either an experiment or a longitudinal study. Experimental methods are superior for testing causal models. However, TPs are fairly stable attributes of personality [1,2] and there is a lack of validated interventions for changing the specific TP dimensions. Thus, we utilized a longitudinal study design for model-testing, given that the experimental approach is not currently feasible.

We aimed to (1) provide an insight into the direction(s) of causality between TPs and WBBs, and to (2) test whether the two types of constructs exert causal effects on subjective well-being.

### 1.1. Time Perspectives

The concept of TP can be traced back to early works of Lewin [7] who defined it as “the totality of the individual’s views of his psychological future and psychological pastexisting in a given time” (p. 75). His ideas were further developed and broadened by multiple researchers (see [2], for a review). Over the last two decades the approach endorsed by Zimbardo and Boyd [1,8] has become the most influential and most commonly cited. The authors emphasized the key regulatory role of the ongoing process of temporal framing of individual and social experiences and proposed five basic temporal features: Past-Negative (depicting past ruminations and focus on traumas and regrets), Past-Positive (reflecting a warm, sentimental view of the past), Present-Hedonistic (manifested in striving for immediate pleasures and a short-term focus), Present-Fatalistic (reflected in a helpless focus on here and now resulting from a sense of the lack of control over the past and the future), and Future (a general focus on future goals). These dimensions reflect individual tendencies to use and overuse particular “temporal horizons”. Thus, in their approach, TP has been defined as a dynamic cognitive process but measured as a set of relatively stable individual differences. These apparent conceptual inconsistencies and resulting problems in understanding TP have been addressed by Stolarski, Fieulaine, and Zimbardo [2] who explicitly distinguished between the momentary focus on one of the temporal horizons, referred to as state-TP, and stable dispositional tendencies to focus on or neglect particular temporal perspectives, labeled trait-TPs. Beside the five dimensions initially identified by Zimbardo and Boyd [1], the universe of TPs has been consequently broadened with such features as Present-Eudaimonic [9], Carpe Diem [10], or Future-Negative [11]. In the present study we used the six-dimensional approach to TP endorsed by Carelli and colleagues [11]. The Swedish authors distinguished between Future-Positive (viewing the future in terms of opportunities, focus on attaining personal goals and consideration of future consequences of current behavior) and Future-Negative (reflected in an anxious attitude towards the future and focus on threats) dimensions and adopted the four past and present perspectives introduced by Zimbardo and Boyd [1]. Both the particular TP dimensions and a joint indicator of temporal adaptation, referred to as balanced TP (e.g., [12]), have been consequently linked to subjective well-being in numerous studies, often explaining incremental portion of variance over and above well-established predictors of happiness and well-being, such as the Big Five personality traits (e.g., [13,14]. However, the mechanisms underpinning this association remain speculative.

### 1.2. Well-Being and Well-Being Boosters

Well-being is one of the central concepts in contemporary psychology; however, its definitions and operationalizations differ between various conceptual approaches (e.g., [15,16]). In general, we may distinguish two major traditions in thinking about happiness and well-being: hedonic and eudaimonic [17]. Proponents of the former approach tend to characterize well-being in terms of affective balance and strive for life satisfaction [15]. Advocates of the eudaimonic approach emphasize the role of psychological strengths and virtues, and focus on promotion of engaged and meaningful life [18]. In the present article we refer to the subjective well-being perspective (SWB) that focuses on the hedonic aspect of well-being. The decision to focus solely on hedonic well-being stemmed from the fact that a vast majority of the research on the associations between TPs and well-being conducted to date referred exclusively to this aspect of well-being. As we sought to further explore the mechanisms of the already well-established effects, and not to explore new research pathways, we did not seek to investigate eudaimonic features of well-being.

Analyses of happiness and well-being seek to identify behavioral and cognitive-affective activities that have positive impacts [19]. That is, well-being is not simply a fixed personality trait; it can be enhanced by deliberate participation in positive activities. WBBs are an example of activities that people can engage in voluntarily. They are thus more malleable than personality traits.

Gratitude refers to people’s tendency to recognize and acknowledge positive experiences and outcomes and their perceived sources [20]. A great number of studies has emerged suggesting that gratitude is strongly related to subjective well-being, as well as to time perspectives [21,22,23]. Gratitude commonly occurs when people retrieve memories that they have encountered a favorable circumstance. Perceptions of favorable circumstances and benefits may be influenced by perceptions of the past, which links gratitude with Past-Positive and Past-Negative TPs.

Savoring reflects reminiscence, self-immersion or anticipation of positive experiences [24]. The savoring process involves noticing and attending to a positive experience, and interpreting and responding cognitively or behaviorally to this stimulus with a savoring response or strategy. Savoring beliefs include three temporal components: past, present, and future. Savoring was found to be associated with greater life satisfaction, positive affect, and more frequent happiness [25], as well as decreased depression and negative affect [26]. In the present article we focus on the most commonly studied and cited aspect of savoring, i.e., savoring the present moment which refers to intensifying or prolonging positive feelings through specific thoughts and behaviors during a pleasant event. The focus on present moment and explicit reference to enhancing momentary pleasure in an active, purposeful way, suggest a clear connotation to present temporal focus which should be reflected in positive links with Present-Hedonistic and inverse associations with Present-Fatalistic.

Prioritizing positivity is defined as “the extent to which individuals seek out positivity by virtue of how they make decisions about how to organize their day-to-day lives” (p. 1159) [27]. By making conscious, purposeful choices regarding activities involving positive emotions, individuals maximize the likelihood of spontaneously experiencing positive emotions in day-to-day life. Prioritizing positivity has a positive correlation with positive emotions and life satisfaction as well as a negative correlation with negative emotions and depression [27]. Given the necessity of deliberate planning of the activities enhancing subjective well-being, prioritizing positivity clearly requires proactivity; hence, it can be considered a future-oriented WBB.

### 1.3. Linking Time Perspectives, Well-Being Boosters, and Subjective Well-Being

As we already mentioned, Burzynska and Stolarski [5], in their attempt to better understand the nature of the links between TPs and well-being, focused on the temporal correspondence between the “positive” TP dimensions and the three WBBs. Out of their four initial models, three still await empirical investigation. In the present paper we report the results of a seminal attempt to answer the question, which of the models (if any) finds the most convincing support in our data collected in a two-wave, one-year longitudinal study. Below, we briefly describe each of the conceptual models.

#### 1.3.1. Trait-Behavior Model

The trait-behavior model stems from the fact that TPs have vital regulatory consequences for the person’s behaviors (see [2]) as well as for intention-behavior consistency [28]. The impact of positive activities on well-being depends in part on person-activity fit, i.e., the individual’s motivation and competence in performing the activity [19,29]. Traits may influence the overall efficacy of the activity in enhancing well-being. Thus, the model suggests that trait-TPs might influence the frequency of engaging in various WBB practices, specific for the defining temporal horizon of the particular TP, indirectly leading to increased or lowered well-being [5]. For example, Past-Positive TP would plausibly increase the probability of practicing and experiencing gratitude, which, in turn, would positively impact well-being. TPs might also moderate the strength of the impact of WBBs. For example, a highly future-oriented person might not experience much of an emotional uplift by reflecting on pleasant past events.

#### 1.3.2. Accumulation Model

The accumulation model proposes a reverse causal chain. It proposes that a systematic exercise of a particular WBB may lead to a gradual change in trait-TPs which, in consequence, results in changes in mental well-being. For instance, practicing savoring activities repeatedly may enhance the respective temporal focus (i.e., the Present-Hedonistic TP), indirectly leading to an elevation in mental well-being. The model is inconsistent with typical assumptions that traits, including TPs, are stable in adulthood [30]. However, recent personality research suggests that traits are more malleable over the lifespan than is traditionally assumed [31]. Furthermore, Zimbardo and Boyd’s [8] account of TPs emphasizes on their sensitivity to environmental factors such as the family and cultural milieu, implying that TPs may be more sensitive to the person’s choice of activities than are biologically based temperamental traits.

#### 1.3.3. Feedback-Loop Model

The last model, referred to as the feedback loop model, assumes a mutual relationship between WBBs and TPs: it proposes that gratitude, savoring the moment, prioritizing positivity, and respective positive TPs strengthen one another, and that such a reciprocal influence contributes to higher mental well-being. To be more specific, one’s stable, positive TP profile influences the probability of undertaking certain WBBs. At the same time, practicing a particular WBB that results in repeated mental “visits” within the respective temporal horizon might reinforce the TP associated with the WBB. Thus, there could be a virtuous cycle that over time enhances well-being and both its TP and WBB supports [32]. An analogical, but reverse pattern is predicted for the “negative” TP dimensions. For example, Past-Negative may diminish gratitude practice, which in turn could further enhance the negative view of the past. The feedback-loop model is thus a superposition of the trait-behavior and accumulation models, and would be supported if both the cross-lagged paths between a TP dimension and the respective WBB are significant.

### 1.4. The Present Study

The present study is part of an ongoing project on the role of TPs in well-being. Burzynska-Tatjewska and colleagues [6] reported initial findings from the project based on two community samples recruited in Poland. At this stage, only cross-sectional data were available. Given this constraint, the study reported tests only of the match-mismatch model because it predicted interaction effects that could be tested in the study samples. The present study builds on this prior work by reporting a longitudinal study capable of testing predictions from the additional models specified by Burzynska and Stolarski [5]. A subset of the participants in Burzynska-Tatjewska et al.’s study [6] were recruited to provide a second wave of data to support additional model testing (see Method for additional details).

The background to the study is provided by cross-sectional studies showing that well-being measures are associated with both TPs and WBBs (see [22,33,34]). For instance, savoring was demonstrated to positively correlate with higher levels of life satisfaction [35] and well-being [36,37]. Similarly, prioritizing positivity was positively associated with subjective happiness, life satisfaction, and psychological well-being [38]. Gratitude too seems to be reliably associated with higher life satisfaction [39]. Turning to TPs, a recent meta-analysis showed that positively anchored TPs (Past-Positive, Present-Hedonistic and Future) were positively associated with indicators of higher subjective well-being such as life satisfaction, positive affect, and negatively linked to indicators of lower well-being such as negative affect and depressive symptoms [40]. Analogously, negatively anchored TPs (Past-Negative, Present-Fatalistic) were negatively related to all positive indicators of well-being and positively associated with negative indicators of well-being.

However, the interplay between TPs and WBBs has been neglected. One exception is a demonstration that gratitude mediates the relationship between TPs and life [22], a finding consistent with the trait-behavior model [5]. This study is limited by its cross-sectional design. Our initial wave of data collection [6] is still the most thorough investigation of the issue, although also cross-sectional in nature. Consistent with the results of the studies just reviewed, all the TP dimensions and WBBs were significantly related to well-being, in predicted directions. The study also tested whether WBBs were associated with different time horizons as expected. The temporal anchoring of gratitude was quite strongly supported in the study; both past TPs were markedly associated with gratitude (0.46 and −0.44, respectively for Past-Positive and Past-Negative). In addition, Present-Fatalistic and Present-Hedonistic were significantly correlated with savoring (−0.34 and 0.12), as anticipated. Surprisingly, the study found little, if any, evidence for the hypothesized links between future-oriented TPs and prioritizing positivity (−0.10 and 0.07, respectively for Future-Negative and Future-Positive, with the latter effect not reaching the significance threshold). In fact, Present-Hedonistic proved more strongly associated with prioritizing positivity. As already mentioned, the study also reported the novel finding that TPs and WBB have interactive effects on well-being, although interactions were generally incompatible with Burzynska and Stolarski’s [5] match-mismatch model.

In this article, we present a one-year longitudinal study with self-report gathered in two waves. We aimed to provide a seminal test of the directions of causality in the TP-WBB associations. Hence, we formulated the following hypotheses based on the predictions of the trait-behavior model:

**Hypothesis** **1** **(H1).***There is a significant cross-lagged effect of (a) Past-Positive (positive effects) and (b) Past-Negative (negative effects) TPs on gratitude and well-being. (c) There is a significant cross-lagged effect of gratitude on well-being*.

**Hypothesis** **2** **(H2).***There is a significant cross-lagged effect of (a) Present-Hedonistic (positive effect) and (b) Present-Fatalistic (negative effect) perspectives on savoring the moment and well-being. Moreover, (c) there is a significant cross-lagged effect of savoring on well-being*.

**Hypothesis** **3** **(H3).***There is a significant cross-lagged effect of (a) Future-Positive (positive effect) and (b) Future-Negative (negative effect) perspectives on prioritizing positivity and well-being. Moreover, (c) there is a significant cross-lagged effect of prioritizing positivity on well-being*.

To test the accumulation model, we proposed the following hypotheses, predicting the opposite direction of causality to those formulated within the trait-behavior model:

**Hypothesis** **4** **(H4).***There is a significant cross-lagged effect of gratitude on (a) Past-Positive (positive effects) and (b) Past-Negative (negative effects) perspectives and well-being*.

**Hypothesis** **5** **(H5).***There is a significant cross-lagged effect of savoring the moment on (a) Present-Hedonistic (positive effects) and (b) Present-Fatalistic (negative effects) perspective and well-being*.

**Hypothesis** **6** **(H6).***There is a significant cross-lagged effect of prioritizing positivity on (a) Future-Positive (positive effects) and (b) Future-Negative (negative effects) perspective and well-being*.

Based on assumptions of the feedback-loop model, we hypothesized that hypotheses 1 and 4 (H7), 2 and 5 (H8), as well 3 and 6 (H9) are simultaneously true.

## 2. Materials and Methods

### 2.1. Participants and Procedures

The first wave of the study was administered in spring of 2019, and the second wave was conducted a year later, already during the COVID-19 pandemic. The initial sample comprised 451 adults (79% women) living in Poland. That sample was used for the test of the match-mismatch model reported in the paper by Burzynska-Tatjewska and colleagues [6] (study 1). For the present analyses, we took into account only the data from participants who completed both measurements. The ultimate sample comprised 206 adults (74% women). During the first wave, the participants’ age ranged from 18 to 63 (M = 30.90, SD = 8.39); during the second wave they were respectively one year older. The participants were approached through social media platforms. Prior to completing the questionnaires, all participants gave their informed consent to take part in a longitudinal study on well-being and time perspectives. Next, they filled in six questionnaires measuring time perspectives, gratitude, savoring the moment, prioritizing positivity, satisfaction with life, positive and negative affect. The study was confidential: the participants provided their email address while completing the online questionnaires. It enabled combining the psychometric data from the first and the second wave of the study.

Both waves of the study were conducted online, using the Qualtrics platform. The participants agreed to take part in the study voluntarily and received no compensation. The online website required an answer to each item; hence, there were no missing data. All individuals were offered feedback on the general results of the study. The entire procedure was approved by the Institutional Ethics Committee of the SWPS University of Social Sciences and Humanities.

### 2.2. Measures

Zimbardo Time Perspective Inventory (ZTPI) [1] measures five TP dimensions: Past-Negative, Present-Hedonistic, Past-Positive, Present-Fatalistic, and Future. We included eight Future-Negative items in our research, as suggested by Carelli et al. [11]. Respondents rated each statement on a five-point Likert scale from “1 = very un-characteristic” to “5 = very characteristic”. Jochemczyk and colleagues [41] reported the extended ZTPI version’s appropriate reliability in the Polish population (Cronbach’s α = 0.86 for Past-Negative, α = 0.70 for Past-Positive, α = 0.82 for Present-Hedonistic, α = 0.75 for Present-Fatalistic, α = 0.76 for Future-Positive, and α = 0.71 for Future-Negative TP). They also confirmed the superior fit of the six-factor model of TP over the original five-factor model endorsed by Zimbardo and Boyd [1].

Gratitude Questionnaire (GQ6) [20] measures the disposition to experience gratitude in everyday life. It contains six items which individuals validate on a seven-point Likert-type scale (1 = strongly disagree, 7 = strongly agree). As Kossakowska and Kwiatek [42] showed, the reliability (Cronbach’s α = 0.71) and validity (correlations with life satisfaction and various conceptually related personality traits) of the Polish version of the scale are satisfactory.

Savoring Beliefs Inventory (SBI) [24] consists of 24 items. It estimates perceived capacity to savor pleasant situations. The questionnaire assesses three aspects of savoring: savoring future events, reminiscing savoring, and savoring the present experience. The savoring the moment subscale was employed in this study. It consists of eight items, each of which is assessed on a seven-point Likert scale (1 = strongly disagree; 7 = strongly agree). Bryant [24] claimed that it has reasonably good reliability (Cronbach’s α coefficients ranging from 0.68 to 0.89). The present subscale of SBI, applied in this study, proved valid also in Polish language version, showing marked correlations with indicators of well-being as well as high reliability (α = 0.87) [6].

Prioritizing Positivity Scale [43] evaluates the level to which individuals look for positive emotional events when planning daily life. The scale contains 6 items that are rated on the nine-point Likert-type scale (1 = disagree strongly, 9 = agree strongly). Catalino and Boulton [43] stated that it has acceptable reliability (omega total coefficient ranging from 0.79 to 0.82) and that it is valid.

#### Subjective Well-Being

Diener et al. [15] (see also [44]) proposed that there are three components to SWB: positive affect, negative affect, and life satisfaction. Following that model, we used Principal Axis Factoring to undertake an exploratory factor analysis to find a higher-order indicator of well-being. Three variables were added: life satisfaction, positive affect, and negative affect, all of which were measured using the scales outlined below. The analysis revealed a single component with an Eigenvalue of 1.82 that accounted for 60.54 percent of the variance among the dimensions in the first wave. The factor loadings were: satisfaction with life (0.88), positive affect (0.60), negative affect (−0.46). For the second wave, the analysis yielded a first factor with an Eigenvalue of 1.74 that accounted for 57.90 percent of the variance among the dimensions. The factor loadings were: satisfaction with life (0.73), positive affect (0.56), and negative affect (−0.54).

Positive and Negative Affect Scale (PANAS) [45] is a self-report measure of mood in the current moment. It consists of 10 positive and 10 negative mood items calculated separately on a five-point Likert-type scale from 1 = very slightly or not at all to 5 = extremely, with higher scores indicating greater positive (or negative) state mood. The sample items are “excited” for positive mood and “upset” for negative mood. Watson and colleagues [45] report good internal consistency of the scales (α = 0.87 for PA, α = 0.85 for NA).

The Satisfaction with Life Scale (SWLS), [46] measures respondent’s satisfaction with life as a whole. It consists of 5 items that are rated on a 7-point Likert-type scale from 1 = strongly disagree; to 7 = strongly agree. Jankowski [47] showed that the Polish version of the scale has adequate reliability (Cronbach’s α = 0.86).

### 2.3. Statistical Analyses

All statistical analyses were conducted using SPSS 27 for Windows supplemented with AMOS 27 for Windows. Preliminary analyses were conducted using Pearson’s correlations and dependent *t*-tests. The main analyses included cross-lagged panel models [48].

## 3. Results

### 3.1. Preliminary Analyses

Given the longitudinal character of the present study, we first tested for changes in study variable means between the first and the second measurement. It should be appreciated that the first wave of the study was conducted before the COVID pandemic (in the April/May 2019), while the second wave of data collection was approximately two months into the pandemic (the first case of COVID 19 in Poland was reported on 4 March 2020). At this time, the levels of anxiety and stress due to the increasing number of cases were already high and the lockdown restrictions were stringent, including the closure of educational institutions, childcare facilities, and all non-essential services (see [49] for a detailed description of all lockdown regulations introduced in Poland in that period). The series of dependent *t*-tests (see Appendix A
Table A1) showed no significant differences between the two waves, except for Present-Hedonistic, Past-Negative TPs, and well-being. However, when Bonferroni correction was applied, none of these effect remained statistically significant. The observed shift in well-being may reflect the overall response to the threatening situation of pandemic. However, the magnitude of these effects did not exceed the level of d = 0.12 which makes them rather negligible (far below d = 0.20, Cohen’s [50] standard for a small effect size).

Next, we tested for selective attrition. The series of independent *t*-tests showed statistically significant differences for Past-Positive, Present-Fatalistic, Future-Positive TPs, and prioritizing positivity; however, the effects are again small (see Appendix B
Table A2), with the greatest difference observed for Future-Positive amounting to d = 0.23. The effect is consistent with the data reported by Harber, Zimbardo and Boyd [51], who showed that TP affects compliance with the demands of longitudinal research. Nevertheless, after Bonferroni correction for multiple comparisons, only the difference in well-being remained significant. Given the small magnitude of the effect (see [50]) we did not attempt to apply any of the data treatment strategies aiming to counter selective attrition.

### 3.2. Correlations

To provide an overview of the associations between WBBs, TPs, and well-being, we calculated Pearson’s correlations. Descriptive statistics and the matrix of correlations between variables measured in both waves of the study are provided in Table 1.

In line with previous results, both in the first and in the second wave, gratitude proved significantly correlated with greater Past-Positive TP and lower Past-Negative TP. Moreover, in both waves savoring the moment was negatively correlated with Present-Fatalistic TP and positively correlated with Present-Hedonistic TP. Additionally, in both waves Present-Hedonistic TP displayed a marked positive correlation with prioritizing positivity. In both waves, there was no significant association between Future-Positive TP and prioritizing positivity, as well as between Future-Negative TP and prioritizing positivity. However, Future-Negative TP displayed a marked negative correlation with savoring the moment. Interestingly, in both waves subjective well-being was associated in predicted directions with all TPs except for Present-Hedonistic in wave 1. The pattern of associations generally remained unchanged when age and gender were controlled for (see Table 1, coefficients above the diagonal).

### 3.3. Cross-Lagged Panel Analyses

Next, we ran a series of cross-lagged panel analyses to estimate the causal effect of WBBs/TPs on well-being. Each of the models was tested in two versions—baseline and with age and gender introduced as control variables (presented in brackets).

The cross-lagged analysis of Past-Positive TP, gratitude and well-being (see Figure 1) was conducted using structural equation modeling. Stability pathways oscillated between 0.57 for gratitude and 0.82 for Past-Positive. The cross-lagged effects proved significant only for Past-Positive TP and gratitude. Thus, we found evidence for the directional effect of Past-Positive on gratitude; however, neither the former nor the latter displayed a significant cross-lagged effect on well-being.

In the case of cross-lagged analysis for Past-Negative TP, gratitude and well-being (see Figure 2), the stability correlations were similar to those obtained in the previous model, between 0.57 for well-being and 0.73 for Past-Negative. The analysis showed the significant cross-lagged effect of Past-Negative TP on well-being, while no opposite cross-lagged effect was observed. No significant causal effects were observed for gratitude in this model.

Then, the cross-lagged analysis of Present-Hedonistic, savoring the moment and well-being was run (see Figure 3). Stability relationships ranged between 0.64 for well-being and 0.81 for savoring the moment. Despite the strong synchronous correlations, there were no significant cross-lagged effects.

Next, we conducted the cross-lagged analysis of Present-Fatalistic TP, savoring the moment and well-being (see Figure 4). Stability relationships oscillated between 0.60 for well-being and 0.80 for savoring. The cross-lagged effects of Present-Fatalistic TP on well-being, as well as Present-Fatalistic TP on savoring the moment proved significant.

Moreover, the cross-lagged analysis of Future-Positive TP, prioritizing positivity and well-being was conducted (Figure 5). Stability relationships were between 0.65 for well-being and 0.78 for Future-Positive. There were no significant cross-lagged effects.

Finally, we ran an analogical cross-lagged analysis for Future-Negative TP, prioritizing positivity and well-being (Figure 6). Stability relationships oscillated around the value of 0.66. A significant cross-lagged effect of prioritizing positivity on Future-Negative TP was shown, suggesting a negative causal effect of the former on the latter. However, when age and gender were controlled, the effect was no longer significant.

## 4. Discussion

The present study aimed to provide an empirical test of the three conceptual models reflecting the possible patterns of associations between TPs, WBBs, and well-being. The hypothesized cross-sectional associations between the TPs, WBBs and mental well-being were largely confirmed. We found that TPs involving negative attitudes towards particular time horizons were related to lower levels of WBBs and lower mental well-being, whereas “positive” TPs were linked to higher levels of WBBs and elevated mental well-being. These findings are largely consistent with previous studies reporting the relationships between TP and subjective well-being (e.g., [13,14]). On the other hand, the claim that particular WBBs can be characterized with a built-in temporal reference found unambiguous evidence only in the case of past TPs and gratitude. Both Past-Positive and Past-Negative were markedly associated with gratitude in both waves of the study (see Table 1), varying around the value of 0.40. Moreover, these correlations were markedly higher than those between gratitude and both present and future TPs. The time 1 associations were already reported by Burzynska-Tatjewska et al. [6]; the current study shows that they remained stable at time 2.

Although present TPs proved associated with savoring, which, at least at the theoretical level, remains a present-oriented WBB, the effects were very weak (ranging between 0.14 and 0.20), and clearly smaller compared with the links between savoring and both past-oriented TPs as well as Future Negative, for which the magnitude of the correlations typically ranged between 0.40 and 0.50. Both Past-Negative and Future-Negative TPs can be understood as sources of distraction, absorbing cognitive resources and impeding present focus, necessary for an in-depth savoring experience [52]. Thus, these associations are not necessarily contrary to the assumptions regarding the temporal anchoring of savoring made by Burzynska and Stolarski [5]. Nevertheless, the weak links between savoring and present-focused TPs, particularly Present-Hedonistic, may suggest that the six-factor model of TPs used in the present study is not sufficient to capture the present nature of savoring, and should be supplemented, for instance, with the Present-Eudaimonistic dimension [9] which depicts a clearly positive present focus. The two present TPs introduced by Zimbardo and Boyd [1] clearly seem to neglect a fundamental role of the positive present focus, manifested in such psychological phenomena as mindfulness or flow state (see also [2]). Here, Present-Fatalistic is clearly maladaptive, whereas Present-Hedonistic remains a complex dimension which, despite some moderate positive effects on such outcomes as well-being or curiosity, may lead to a number of undesirable consequences including risk taking [41] or unhealthy lifestyle [53].

Similarly, we found no evidence for the cross-sectional associations between prioritizing positivity and future-oriented TPs. Instead, the seemingly future-oriented WBB turned out to be related to greater Present-Hedonistic and Past-Positive TPs. The former result seems to reflect the motivational component of Present-Hedonistic, reflected in the strive for pleasure and seeking for intense sensations. Paradoxically, Present-Hedonistic seems to be more a short-term future-oriented dimension (which is reflected in its marked associations with the CFC-Immediate subscale of Consideration of Future Consequences scale; see, e.g., [54]) than a tendency to remain focused on the present moment. Hence its stronger associations with prioritizing positivity than with savoring and its boosting interaction with the former dimension are reported by Burzynska-Tatjewska et al. [6]. On the other hand, the link between Past-Positive and prioritizing positivity seems in line with the results of a study where the positive attitude of the past was related to anticipation of positive moods in the future [55]. Importantly, all these effects remained unchanged when age and gender were controlled for.

Apart from the conclusions derived from cross-sectional, correlational analysis, our primary goal was to provide some answers regarding the direction of causality in the links between TPs, WBBs, and subjective well-being. Before we begin to discuss the results of the cross-lagged panel analysis, it is essential to emphasize that the onset of the COVID-19 pandemic took place approximately two months before collecting the data from the second wave. During this period, COVID-19 pandemic was developing in nearly all countries of the world [56]. The COVID-19 pandemic is an extraordinary circumstance and its psychological consequences have been thoroughly explored by multiple scientists (for instance: [57,58]). As a result, this study inadvertently became a natural quasi-experiment, and the observed effects may in fact reflect a process of adaptation to the novel, threatening situation. Obviously, that could have influenced the result presented here. On the other hand, such a situation could provide an opportunity to observe the regulatory consequences of both TPs and WBBs for maintaining well-being in the context of the pandemic. The decline in well-being observed following the onset of the pandemic is consistent with other studies [59]. The small magnitude of the effect may reflect perceptions early in the pandemic that it would be short-lived. In a longitudinal study in Norway, Hansen et al. [60] observed only small declines in well-being relative to pre-pandemic levels in June 2020 but larger effect sizes in November-December 2020 as it became apparent pandemic impacts would not quickly be resolved.

Next, we consider the tests against longitudinal data of each of the three models from Burzynska and Stolarski [5]. On the basis of the trait-behavior model, it was hypothesized that that Past TPs at Time 1 would predict WBBs and well-being at Time 2 in the cross-lagged analyses (H1a, H1b). This hypothesis was partially supported. Past-Positive had a significant cross-lagged effect on gratitude but not well-being, such that Time 1 Past-Positive predicted higher gratitude at Time 2. By contrast, initial Past-Negative appeared to influence later well-being negatively but not gratitude. The effect on well-being is consistent with previous reports that negative and aversive views of the past were inversely related to positive affect [61] and life satisfaction [13]. In previous studies gratitude was found to mediate the relationship between Past-Positive perspective and life satisfaction [22,23]. However, in the present data cross-lagged effects on gratitude and well-being were dissociated, and we could not confirm a role for gratitude in driving future well-being (H1c). Thus, while the cross-lagged effects of past TP are consistent with the trait-behavior model, the role of gratitude in well-being remains somewhat unclear.

Analyses of the Present TPs showed cross-lagged effects for Present-Fatalistic but not Present-Hedonic, partially supporting H2. Present-Fatalistic displayed a significant cross-lagged effect on savoring the moment, consistent with H2b. It appears that a helpless attitude toward current events may disrupt the ability to recognize and cherish positive emotions in the present, whereas individuals with low levels of Present-Fatalistic TP further developed their savoring skills. Time 1 Present-Fatalistic also predicted lower Time 2 well-being, consistent with previous findings from cross-sectional studies [62]. However, we could not confirm a cross-lagged relationship between savoring and later well-being (H2c).

The anticipated effects of Future TPs on prioritizing positivity and well-being (H3a, H3b) were not supported. Future TPs at Time 1 had no significant effects on these outcomes at Time 2. In addition, Time 1 prioritizing positivity was not predictive of Time 2 well-being (H3c).

While some predictions from the trait-behavior model were supported, hypotheses derived from the accumulation model were generally disconfirmed (H4–H6). None of the Time 1 WBBs predicted Time TPs in the expected direction in any of the cross-lagged analyses. Prioritizing positivity displayed a significant, cross-lagged effect on the Future-Negative perspective. However, surprisingly, the effect was positive: individuals with a greater tendency to prioritize positive life events and pleasant activities experienced an increase in the tendency to view the future in a negative way. Speculatively, this outcome might reflect the impact of the pandemic; perhaps the emerging barriers to engagement in positive activities posed by lockdowns and pandemic restrictions highlighted the likelihood of future disappointment. Nevertheless, the effect was no longer significant when age and gender were controlled for. In any case, this finding clearly undermines the assumptions of the accumulation model which states that regular practice of WBBs should lead to the development of a more desirable TP profile (e.g., lower Future-Negative TP). Stolarski and Matthews [14] showed that Future-Negative is the strongest correlate of negative moods and decreased well-being from all TPs. However, again, no causal effects on well-being were observed in that model. Therefore, even if prioritizing positivity may enhance the level of Future-Negative, we found no evidence that any of these variables truly impacts well-being. Combined with the fact that we have found no effects of WBBs on any other TP dimension, we may say that the accumulation model found literally no support in the present study. The lack of anticipated cross-lagged effects of WBBs on TPs also disconfirms the feedback-loop model (H7–H9).

A surprising feature of the data was the lack of evidence for cross-lagged effects of Time 1 WBBs on Time 2 well-being. This finding is surprising given considerable research literature in positive psychology that argues for beneficial effects of these positive activities [19,63]. For example, savoring positive experiences acts as a mediating process between mindfulness and life satisfaction [64], and a one-time communication savoring intervention can promote higher levels of happiness and life satisfaction [35]. One possible explanation is that WBBs have slowing-acting impacts that take more than a year to enhance well-being. Alternatively, WBBs and satisfaction may be reciprocally linked over shorter time frames than the 1-year of the current study. That is, consistent with the cross-sectional correlations, WBBs may produce rapid enhancement of satisfaction that feeds back into maintenance of the WBBs, consistent with a virtuous cycle [63]. Such a dynamic cycle would not necessarily produce the lagged effects tested here. By contrast, impacts of TPs on positive activities and well-being may reflect longer-term adaptative behaviors that take substantial time to produce beneficial outcomes.

## 5. Limitations

The present study has some limitations. Among them, probably the most obvious refers to the methodology for testing causal models. Although longitudinal studies are clearly superior over cross-sectional research, they are still susceptible to such problems as third variable effects, multiple testing, selective attrition or changing historical context, among others [65]. All of these issues might have influenced the present results.

Moreover, it seems it would be a good idea to add psychological well-being measures to our research. In the present paper we examine subjective well-being, that refers to people’s evaluation of their own lives and things which are important to them [66]. In that understanding, happiness consists of the following sub-components—satisfaction with life, a positive evaluation of the most important spheres of one’s life, as well as of frequently experiencing positive emotions and rarely experiencing negative emotions. On the other hand, psychological well-being embraces six dimensions: autonomy, the ability to manage complex environments to suit personal needs and values, personal growth and development, quality ties to others, self-acceptance and pursuit of meaningful goals [67]. Keyes, Shmotkin, and Ryff [16] demonstrated that subjective well-being and mental well-being are related yet separate constructs—as they have distinct socio-demographic and personality correlates, each maintains its uniqueness as a distinct facet of overall well-being. Using measures to examine psychological well-being would be recommendable to better understand the relationships between TPs, WBBs, and well-being and to see if WBBs play a similar role as mediators between TPs and psychological well-being. For example, personal growth and development seems compatible with Future Positive TPs.

Furthermore, the present study included only two waves of assessment. Such a design provides some information regarding the causality of the analyzed associations; however, to draw conclusions about longitudinal mediation effects which were predicted in the three tested conceptual models [5] a study should comprise at least three measurements. The lack of the third measurement is clearly a drawback of the present study, and future studies may seek to re-test the hypotheses using a three-wave design. Nevertheless, the simple cross-lagged effects reported here are sufficient to suggest that none of the three conceptual models is fully supported. Although there was evidence for cross-lagged effects of TPs on WBBs, we found zero evidence for causal effects of WBBs on SWB. Thus, we cannot substantiate mediation models in which TPP affects WBB, which in turn affects SWB. One of the gaps in the current literature is a lack of evidence on the timespan over which causal impacts on SWB operate. As discussed in the previous section, we cannot exclude either possible fast-acting impacts of WBBs producing relatively rapid elevation of well-being or slow-acting impacts that accumulate over periods of years. Future longitudinal research might not only increase the number of measurement points but also explore modeling of timespans ranging from weeks to years. Researchers seeking to replicate these findings using a three-wave design could also increase the sample size (seeking to capture the more subtle effects).

Moreover, although controlling for age and gender did not influence the results of our study, other variables, such as education or religiosity were not included as control variables. Taking into account the marked associations between TP dimensions and religiosity [68], it seems vital to determine role of this feature in the studied relationships. In future research, it would be also advisable to consider a lifetime perspective in order to replicate the investigated relationships across different age groups and to see whether at different life stages WBBs play a comparable role as mediators between time perspectives and subjective well-being. A longer time interval with multiple waves of data collection would provide further insights into dynamic relationships between TPs, WBBs, and well-being. Moreover, the Polish adaptation of the Prioritizing Positivity Scale was made after we conducted our study [69]. In the future research we could conduct studies based on the scale.

Finally, it is essential to remember that the time frames for the present longitudinal study included the onset of the pandemic; hence, the result reported here should be treated with caution. It is worthwhile to further explore how TPs interplay with WBBs in shaping well-being after the pandemic to find out whether the observed causal effects are specific for the lock-down context.

## 6. Conclusions

In sum, we found evidence consistent with the trait-behavior model for three out of six TPs. Consistent with the temporal frames associated with the TPs, Past-Positive had a cross-lagged effect on gratitude, and Present-Fatalistic had an adverse effect on savoring the moment. Both Present-Fatalistic and Past-Negative perspectives had negative impacts on future well-being. These findings are consistent with the perspective on personality provided by McCrae and Costa’s [70] Five Factor Theory which proposes that the Big Five traits influence characteristic adaptations such as personal strivings that are related dynamically to external events and real-life outcomes. Because TPs are closely related to basic temperaments [71,72], they are akin to the Big Five, whereas the more malleable WBBs are a type of characteristic adaptation. Each TP may influence a range of characteristic associated with the differing temporal focuses for cognition and emotion [73]. For some TPs, these adaptations may include WBBs. For example, the present data suggest that the Present-Fatalistic TP maladaptively erodes the person’s capacity for savoring pleasant events in the present, perhaps because of the cynicism characteristic of this TP [74]. However, the partial support for the trait-behavior model implies that WBBs are only one out of several adaptations that may mediate impacts of TPs on well-being and other cognitive-affective outcomes. For three of the TPs, including both future TPs, no cross-lagged impacts on WBBs were detected here, and other types of adaptation may be more important.

The importance of well-being for public is increasingly accepted [75], but the relevance of individual differences in personality for public health policy has not received sufficient attention [76]. The present data reaffirm the importance of TPs as influences on well-being, consistent with Zimbardo and Boyd’s [8] account of the importance of adaptive TPs for societal flourishing. The data also show that TP may impact activities that support well-being, especially gratitude. In principle, the implementation of interventions that elevate gratitude and other WBBs may contribute to personal and societal well-being [77], consistent with positive psychology perspectives [18,19]. However, over a one-year timespan, we could not substantiate a direct causal effect of WBBs on well-being, implying that more research is necessary to determine the circumstances under which practical interventions based on WBB activities are effective. In short, we need to know how long it takes for activities to impact well-being to adequately evaluate the impacts of interventions.

The current findings also suggest that interventions focused on WBBs are unlikely to directly change individuals’ TPs, which will continue to impact well-being post-intervention. However, the efficacy of interventions may be moderated by the person’s pre-existing TPs. Our previous report [6] demonstrated compensatory effects that imply interventions based on WBBs may be most effective for individuals with maladaptive TPs. Although TPs may be difficult to change, practitioners may be able to tailor interventions to individual vulnerabilities associated with personal time horizon.

## Figures and Tables

**Figure 1 ijerph-19-04776-f001:**
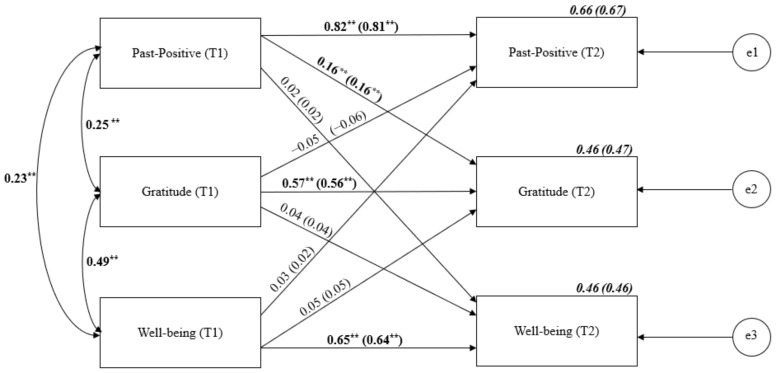
Cross-lagged panel analysis of Past-Positive TP, gratitude and well-being. Notes: Path entries are standardized estimates. T1 and T2 indicate the time when data were collected (Time 1 or Time 2, respectively). Estimates for the corresponding model with age and gender controlled are given in brackets. e1–e3 refers to the residual error or error variance that causes response variation in observed variables. ** *p* < 0.01.

**Figure 2 ijerph-19-04776-f002:**
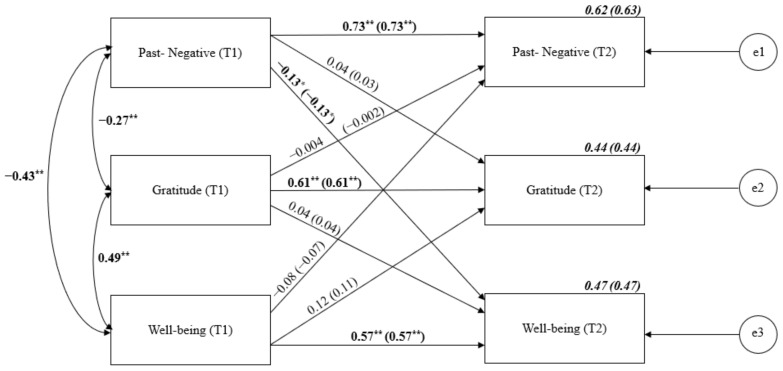
Cross-lagged panel analysis of Past-Negative TP, gratitude and well-being. Notes: Path entries are standardized estimates. T1 and T2 indicate the time when data were collected (Time 1 or Time 2, respectively). Estimates for the corresponding model with age and gender controlled are given in brackets. e1–e3 refers to the residual error or error variance that causes response variation in observed variables. * *p* < 0.05; ** *p* < 0.01.

**Figure 3 ijerph-19-04776-f003:**
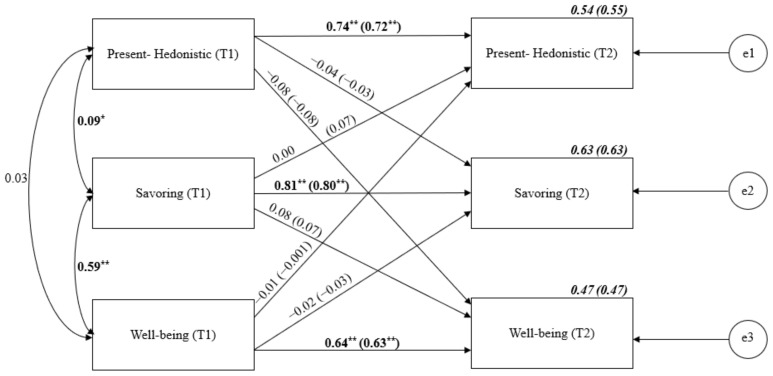
Cross-lagged panel analysis of Present-Hedonistic TP, savoring the moment and well-being. Notes: Path entries are standardized estimates. T1 and T2 indicate the time when data were collected (Time 1 or Time 2, respectively). Estimates for the corresponding model with age and gender controlled are given in brackets. e1–e3 refers to the residual error or error variance that causes response variation in observed variables. * *p* < 0.05; ** *p* < 0.01.

**Figure 4 ijerph-19-04776-f004:**
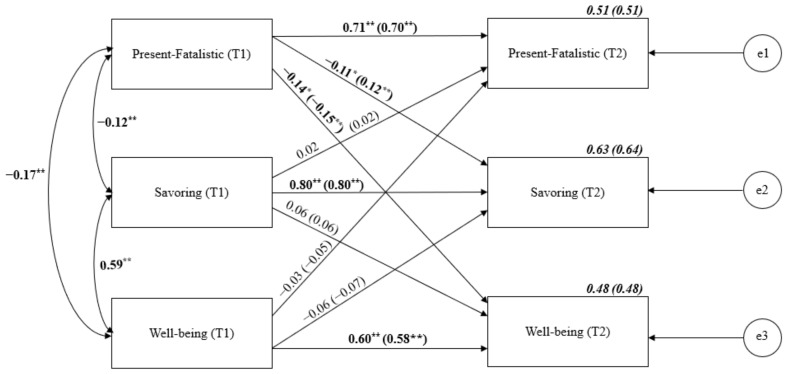
Cross-lagged panel analysis of Present-Fatalistic TP, savoring the moment and well-being. Notes: Path entries are standardized estimates. T1 and T2 indicate the time when data were collected (Time 1 or Time 2, respectively). Estimates for the corresponding model with age and gender controlled are given in brackets. e1–e3 refers to the residual error or error variance that causes response variation in observed variables. * *p* < 0.05; ** *p* < 0.01.

**Figure 5 ijerph-19-04776-f005:**
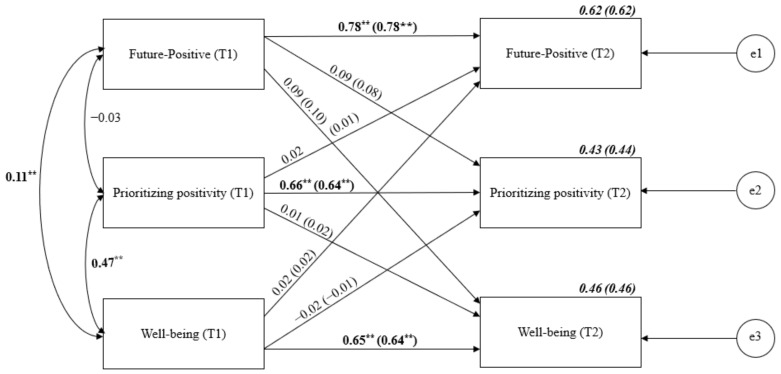
Cross-lagged panel analysis of Future-Positive TP, prioritizing positivity and well-being. Notes: Path entries are standardized estimates. T1 and T2 indicate the time when data were collected (Time 1 or Time 2, respectively). Estimates for the corresponding model with age and gender controlled are given in brackets. e1–e3 refers to the residual error or error variance that causes response variation in observed variables. ** *p* < 0.01.

**Figure 6 ijerph-19-04776-f006:**
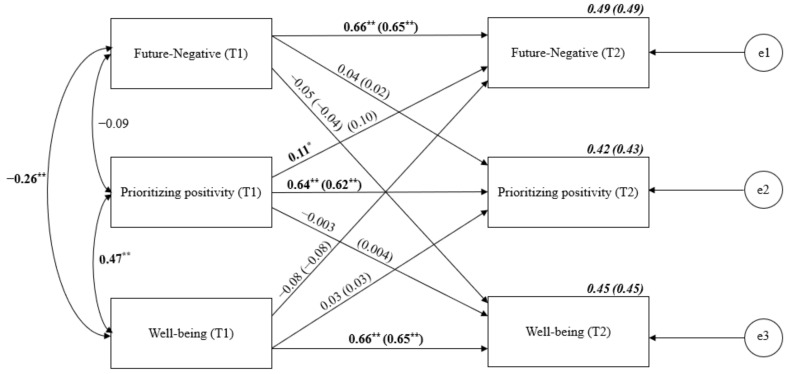
Cross-lagged panel analysis of Future-Negative TP, prioritizing positivity and well-being. Notes: Path entries are standardized estimates. T1 and T2 indicate the time when data were collected (Time 1 or Time 2, respectively). Estimates for the corresponding model with age and gender controlled are given in brackets. e1–e3 refers to the residual error or error variance that causes response variation in observed variables. * *p* < 0.05; ** *p* < 0.01.

**Table 1 ijerph-19-04776-t001:** Means, standard deviations, Cronbach’s alphas, and correlations between variables included in both waves (N = 206).

		*M*	*SD*	*α*	1	2	3	4	5	6	7	8	9	10	11	12	13	14	15	16	17	18	19	20
1	Past-Negative (1)	2.88	0.75	0.84	-	−0.35 **	0.42 **	0.08	0.56 **	−0.14 *	−0.36 **	−0.49 **	−0.16 *	−0.63 **	0.78 **	−0.26 **	0.36 **	0.06	0.46 **	0.17 *	−0.25 **	−0.47 **	−0.09	−0.50 **
2	Past-Positive (1)	3.58	0.63	0.76	−0.37 **	-	−0.09	0.06	−0.15 *	0.09	0.38 **	0.30 **	0.27 **	0.37 **	−0.30 **	0.80 **	−0.03	0.05	−0.10	0.15	0.39 **	0.25 **	0.19 **	0.27 **
3	Present-Fatalistic (1)	2.42	0.60	0.72	0.39 **	−0.06	-	0.32 **	0.43 **	−0.47 **	−0.27 **	−0.22 **	0.07	−0.36 **	0.33 **	−0.10	0.71 **	0.24 **	0.33 **	−0.44 **	−0.25 **	−0.27 **	0.02	−0.37 **
4	Present-Hedonistic (1)	3.24	0.52	0.78	0.07	0.08	0.33 **	-	−0.02	−0.36 **	0.11	0.14 *	0.32 **	0.05	0.13	0.05	0.26 **	0.72 **	−0.01	−0.35 **	0.00	0.08	0.23 **	−0.03
5	Future-Negative (1)	3.04	0.56	0.67	0.56 **	−0.16 *	0.42 **	0.00	-	−0.08	−0.30 **	−0.48 **	−0.14 *	−0.51 **	0.42 **	−0.07	0.30 **	−0.01	0.68 **	0.00	−0.16 *	−0.41 **	−0.08	−0.37 **
6	Future-Positive (1)	3.65	0.51	0.73	−0.15 *	0.12	−0.44 **	−0.32 **	−0.08	-	0.21 **	0.01	−0.05	0.22 **	−0.17 *	0.08	−0.33 **	−0.33 **	−0.08	0.78 **	0.25 **	0.03	0.05	0.24 **
7	Gratitude (1)	5.54	0.98	0.79	−0.38 **	0.41 **	−0.23 **	0.14 *	−0.30 **	0.23 **	-	0.43 **	0.33 **	0.54 **	−0.31 **	0.26 **	0.24 **	0.00	−0.21 **	0.28 **	0.65 **	0.38 **	0.24 **	0.39 **
8	Savoring (1)	4.77	1.10	0.85	−0.50 **	0.33 **	−0.19 **	0.15 *	−0.48 **	0.03	0.45 **	-	0.37 **	0.58 **	−0.38 **	0.23n **	−0.15 *	0.11	−0.34 **	0.01	0.25 **	0.78 **	0.20 **	0.42 **
9	Prioritizing positivity (1)	7.01	1.54	0.85	−0.15 *	0.26 **	0.07	0.34 **	−0.10	−0.04	0.33 **	0.36 **	-	0.35 **	−0.08	0.19 **	0.07	0.25 **	−0.02	−0.02	0.23 **	0.28 **	0.64 **	0.24 **
10	Well-being (1)	3.50	0.90	0.83	−0.65 **	0.40 **	−0.31 **	0.07	−0.51 **	0.24 **	0.56 **	0.60 **	0.34 **	-	−0.54 **	0.29 **	−0.27 **	0.04	−0.38 **	0.20 **	0.41 **	0.44 **	0.24 **	0.66 **
11	Past-Negative (2)	2.80	0.69	0.82	0.79 **	−0.32 **	0.30 **	0.12	0.43 **	−0.18 **	−0.33 **	−0.40 **	−0.06	−0.56 **	-	−0.30 **	0.35 **	0.08	0.46 **	−0.17 *	−0.29 **	−0.45 **	−0.06	−0.58 **
12	Past-Positive (2)	3.56	0.67	0.81	−0.28 **	0.81 **	0.02	0.09	−0.08	0.11	0.30 **	0.27 **	0.20 **	0.33 **	−0.32 **	-	0.04	0.11	−0.08	0.10	0.40 **	0.23 **	0.20 **	0.27 **
13	Present-Fatalistic (2)	2.48	0.57	0.72	0.34 **	−0.01	0.71 **	0.28 **	0.29 **	−0.30 **	−0.21 **	−0.13	0.08	−0.24 **	0.33 **	0.07	-	0.24 **	0.35 **	−0.39 **	−0.25 **	−0.22 **	0.01	−0.37 **
14	Present-Hedonistic (2)	3.19	0.51	0.79	0.06	0.06	0.24 **	0.74 **	0.03	−0.29 **	0.02	0.11	0.29 **	0.05	0.09	0.13	0.25 **	-	−0.10	−0.30 **	0.05	0.16 *	0.31 **	0.18 **
15	Future-Negative (2)	3.04	0.52	0.65	0.47 **	−0.11	0.32 **	0.03	0.69 **	−0.08	−0.20 **	−0.34 **	0.02	−0.38 **	0.47 **	−0.08	0.35 **	−0.04	-	0.02	−0.16 *	−0.40 **	−0.03	−0.42 **
16	Future-Positive (2)	3.66	0.52	0.78	−0.19 **	0.17 *	−0.41 **	−0.31 **	0.00	0.79 **	0.30 **	0.03	0.00	0.22 **	−0.18 **	0.12	−0.37 **	−0.27 **	0.02	-	0.34 **	0.09	0.11	0.26 **
17	Gratitude (2)	5.47	0.94	0.83	−0.27 **	0.41 **	−0.21 **	0.04	−0.016 *	0.27 **	0.66 **	0.27 **	0.24 **	0.43 **	−0.30 **	0.43 **	−0.21 **	0.08	−0.15 *	0.36 **	-	0.38 **	0.31 **	0.50 **
18	Savoring (2)	4.80	1.05	0.87	−0.49 **	0.28 **	−0.24 **	0.08	−0.42 **	0.05	0.40 **	0.79 **	0.26 **	0.46 **	−0.47 **	0.25 **	−0.20 **	0.14 *	−0.41 **	0.10	0.39 **	-	0.29 **	0.57 **
19	Prioritizing positivity (2)	7.02	1.40	0.84	−0.07	0.18 **	0.02	0.26 **	−0.04	0.06	0.24 **	0.19 **	0.65 **	0.23 **	−0.03	0.21 **	0.02	0.34 **	0.02	0.12	0.32 **	0.26 **	-	0.33 **
20	Well-being (2)	3.48	0.82	0.85	−0.51 **	0.30 **	−0.34 **	−0.02	−0.38 **	0.25 **	0.41 **	0.44 **	0.22 **	0.068 **	−0.59 **	0.30 **	−0.34 **	0.17 *	−0.43 **	0.27 **	0.51 **	0.59 **	0.31 **	-

Note: ** *p* < 0.01, * *p* < 0.05. Bivariate correlations are presented below the diagonal. Partial correlations, controlling for age and gender, are provided above the diagonal. (1)—the first wave of the study, (2)—the second wave of the study.

## Data Availability

The presented data in this study are available from the corresponding author upon reasonable request.

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
