# Peer review of "Seeking Causality in the Links between Time Perspectives and Gratitude, Savoring the Moment and Prioritizing Positivity: Initial Empirical Test of Three Conceptual Models"

_ijerph, 2022, doi:10.3390/ijerph19084776_

Round 1

Reviewer 1 Report

This is an interesting and generally well-written paper examining the causal links between time perspectives, gratitude, savoring the moment, and prioritizing positivity using a cross-lagged panel. The major advantage of the study is the implementation of a longitudinal survey.

Despite my overall positive opinion on this paper, I have some concerns that should be addressed prior to publication:

1) As we work in the falsification framework, we cannot verify anything in our studies. Rather, data can support our theoretical models or hypotheses. Please change the title and relevant sentences in the manuscript (e.g., "We provide an initial empirical verification…, " "That sample was used for the verification of the match-mismatch model…," "The present study aimed to verify the three conceptual models…, " "In the future research, it would be also advisable to consider a lifetime perspective in order to verify the investigated relationships…").

2) Please add the mean age of participants and the proportion of genders to the abstract. Moreover, please add the proportion of genders at the second assessment in the Materials and Methods section.

3) There is little information concerning the participants' characteristics. Did the Authors measure education level and religiosity? Especially religiosity may play a significant role in the investigated relationships.

4) "To address the key role of causal mechanisms in each of the models, such a verification required either an experiment or a longitudinal study." – This sentence may suggest that experimental and longitudinal designs support the hypothesis concerning the causal relationships to the same degree. The limitations of the longitudinal methodology while testing the cause-and-effect relationships should be mentioned in the paper.

5) Why did the Authors decide to focus on hedonic well-being and did not investigate eudaimonic well-being? This should be clearly explained in the Introduction.

6) How did the Authors manage the missing data?

7) It would be helpful to add the experienced variance (R2) to the figures.

8) Did the Authors control the model for sociodemographics?

9) The limitations related to including only two waves of assessment should be mentioned.

10) Proofreading the manuscript before the second round of reviewers' comments is highly recommended.

Author Response

Reviewer #1

This is an interesting and generally well-written paper examining the causal links between time perspectives, gratitude, savoring the moment, and prioritizing positivity using a cross-lagged panel. The major advantage of the study is the implementation of a longitudinal survey.

Despite my overall positive opinion on this paper, I have some concerns that should be addressed prior to publication:

1) As we work in the falsification framework, we cannot verify anything in our studies. Rather, data can support our theoretical models or hypotheses. Please change the title and relevant sentences in the manuscript (e.g., "We provide an initial empirical verification…, " "That sample was used for the verification of the match-mismatch model…," "The present study aimed to verify the three conceptual models…, " "In the future research, it would be also advisable to consider a lifetime perspective in order to verify the investigated relationships…").

Response from Authors:

Thank you for this point. We substituted “verification” with “test” across the entire manuscript. The title and the respective sentences now read as follows:

“Seeking causality in the links between time perspectives and gratitude, savoring the moment and prioritizing positivity: Initial empirical test of three conceptual models”

“We provide an initial empirical test of three conceptual models…”

“However, the results of two studies aiming to test the model”

“To address the key role of causal mechanisms in each of the models, such a test required either…”

“That sample was used for the test of the match-mismatch model…”

“The present study aimed to provide and empirical test of the three conceptual models…”

“In future research, it would be also advisable to consider a lifetime perspective in order to replicate the investigated relationships…”

2) Please add the mean age of participants and the proportion of genders to the abstract. Moreover, please add the proportion of genders at the second assessment in the Materials and Methods section.

Response from Authors:

As suggested, we added the mean age of participants and the proportion of genders to the abstract. Regarding the proportion of genders at the second assessment, it was the same at both assessments (the ultimate sample was exactly the same and comprised the same 206 adults as the subsample of wave 1 which took part in both measurements). It is possible that the Reviewer meant the proportion of genders in the full-sample in wave 1; we added the relevant information for the total sample which filled the questionnaires during the first measurement. This sentence now reads as follows:

“The initial sample comprised 451 adults (79% women) living in Poland. That sample was used for the test of the match-mismatch model reported in the paper by Burzynska-Tatjewska and colleagues [6] (study 1). For the present analyses we took into account only the data from participants who completed both measurements. The ultimate sample comprised 206 adults (74% women).”

3) There is little information concerning the participants' characteristics. Did the Authors measure education level and religiosity? Especially religiosity may play a significant role in the investigated relationships.

The data concerning education level and religiosity were not collected. Thank you for the comment, we will include the measurements in our future studies. We also added this to the limitations section; the relevant part now reads as follows:

“Moreover, although controlling for age and gender did not influence the results of our study, other variables, such as education or religiosity were not included as control variables. Taking into account the marked associations between TP dimensions and religiosity [68] it seems vital to determine role of this feature in the studied relationships.”

  1. Łowicki, P.; Witowska, J.; Zajenkowski, M.; Stolarski, M. Time to believe: Disentangling the complex associations between time perspective and religiosity. Pers. Indiv. Differ. 2018, 134, 97-106. https://doi.org/10.1016/j.paid.2018.06.001

4) "To address the key role of causal mechanisms in each of the models, such a verification required either an experiment or a longitudinal study." – This sentence may suggest that experimental and longitudinal designs support the hypothesis concerning the causal relationships to the same degree. The limitations of the longitudinal methodology while testing the cause-and-effect relationships should be mentioned in the paper.

Response from Authors:

We have revised this part of Introduction – it now reads as follows:
“To address the key role of causal mechanisms in each of the models, such a test required either an experiment or a longitudinal study. Experimental methods are superior for testing causal models. However, TPs are fairly stable attributes of personality [2, 3] and there is a lack of validated interventions for changing the specific TP dimensions. Thus, we utilized a longitudinal study design for model-testing, given that the experimental approach is not currently feasible.”

Moreover, we discussed the limitations of the longitudinal methodology in the Limitations section:

“…Among them, probably the most obvious refer to the methodology for testing causal models. Although longitudinal studies are clearly superior over cross-sectional research, they are still susceptible to such problems as third variable effects, multiple testing, selective attrition or changing historical context, among others [65]. All of these issues might have influenced the present results.”

  1. Alwin D.F.; Campbell R.T. Quantitative approaches. Longitudinal methods in the study of human development and aging, In Handbook of aging and the social sciences (5th ed.), Binstock, R.H. Ed.; Academic Press: San Diego, 2001; 22–43.

5) Why did the Authors decide to focus on hedonic well-being and did not investigate eudaimonic well-being? This should be clearly explained in the Introduction.

We have now provided the following explanation in the section 1.2.:

“The decision to focus solely on hedonic well-being stemmed from the fact that vast majority of the research on the associations between TPs and well-being conducted to date referred exclusively to this aspect of well-being. As we sought to further explore the mechanisms of the already well-established effects, and not to explore new research pathways, we did not seek to investigate eudaimonic features of well-being.”

6) How did the Authors manage the missing data?

Response from Authors:

There were no missing data in the study thanks to the settings of the applied online study form which disabled completing the study without providing responses to all the items. This information is now added to the procedure section.

7) It would be helpful to add the experienced variance (R2) to the figures.

Response from Authors:

R2 values were added to the graphs.

8) Did the Authors control the model for sociodemographics?

Response from Authors:

In the original submission the effects of age and gender were not controlled. As each of the three Reviewers of the paper referred to this issue (i.e., the lack of adding age and gender as control variables in the tested models), we decided to re-run all the analyses with age and gender included as control variables. Thus, in the current version we additionally provide partial correlations with age and gender controlled (see Table 1, values above the diagonal). Moreover, across all the cross-lagged analyses we now present the alternative set of coefficients, derived from the models run with age and gender controlled for (see fig 1-6; values presented in the brackets). Importantly, adding these two control variables did not change any of the relevant results of our study. We have now commented on this issue in the manuscript.

9) The limitations related to including only two waves of assessment should be mentioned.

Response from Authors:

The limitation has been discussed in the limitations section, accompanied with a brief discussion of this issue and future research directions:

“Furthermore, the present study included only two waves of assessment. Such a design provides some information regarding the causality of the analyzed associations; however, to draw conclusions about longitudinal mediation effects which were predicted in the three tested conceptual models [1] a study should comprise at least three measurements. The lack of the third measurement is clearly a drawback of the present study, and future studies may seek to re-test the hypotheses using a three-wave design. Nevertheless, the simple cross-lagged effects reported here are sufficient to suggest that none of the three conceptual models is fully supported. Although there was evidence for cross-lagged effects of TPs on WBBs, we found zero evidence for causal effects of WBBs on SWB. Thus, we cannot substantiate mediation models in which TPP affects WBB, which in turn affects SWB. One of the gaps in the current literature is a lack of evidence on the timespan over which causal impacts on SWB operate. As discussed in the previous section, we cannot exclude either possible fast-acting impacts of WBBs producing relatively rapid elevation of well-being or slow-acting impacts that accumulate over periods of years. Future longitudinal research might not only increase the number of measurement points but also explore modeling of timespans ranging from weeks to years. Researchers seeking to replicate these findings using a three-wave design could also increase the sample size (seeking to capture the more subtle effects).”

10) Proofreading the manuscript before the second round of reviewers' comments is highly recommended.

Response from Authors:

The second Author of the manuscript is a native English speaker. He has now doublechecked the article paying particular attention to the correctness of language.

Reviewer 2 Report

Seeking causality in the links between time perspectives and gratitude, savoring the moment and prioritizing positivity: Initial empirical verification of three conceptual models

The article addresses a study of great interest and social relevance, before and after the COVID-19 pandemic.

The authors make a good presentation of the theoretical aspects involved in the study.

I consider that the development of the Introduction section and the general objective and the hypotheses are appropriate.

In general, the structure and methodology of the article is well organized. However, I suggest somes corrections.

CONTENTS

  1. Materials and Methods

Cross-lagged panel analysis (CLPA) is one of a variety of quasi-experimentaI forms of analysis used by an increasing number of social scientists for making causal inferences.

  1. Results

3.3. Cross-lagged panel analyses

The results obtained are described in each of the figures, except in Figure 6. I consider it necessary to include a brief paragraph describing the results of Figure 6.

  1. Discussion

The preparation of the discussion should be more adjusted to the formulated working hypotheses. It would be easier to trace the arguments adduced to the results obtained. indicate whether or not the hypothesis in question is accepted.

  1. Limitations

The participants' age ranged from 18 to 63. It is possible that the age of the participants and the sex may have influenced the results?

I believe that the authors should include some comment on this.

References                                                                           

- Check the references. Some do not follow the format of the magazine. For example:

  1. Hansenne, M. Valuing Happiness is Not a Good Way of Pursuing Happiness, but Prioritizing Positivity is: A Replication Study. Psychol. Belg. 2021, 61(1), pp. 306–314. doi: https://doi.org/10.5334/pb.1036

Delete “pp”

Delete “doi”

  1. Cohen, J. Statistical Power Analysis. Curr. Dir. Psychol. Sci. 1992;1(3):98-101. doi:10.1111/1467-8721.ep10768783

Delete “ ; ”

To sum up, after explaining the strengths and weaknesses in the text, I recommend publishing the article after the indicated corrections.

Author Response

The article addresses a study of great interest and social relevance, before and after the COVID-19 pandemic.

The authors make a good presentation of the theoretical aspects involved in the study.

I consider that the development of the Introduction section and the general objective and the hypotheses are appropriate.

In general, the structure and methodology of the article is well organized. However, I suggest somes corrections.

CONTENTS

  1. Materials and Methods

Cross-lagged panel analysis (CLPA) is one of a variety of quasi-experimentaI forms of analysis used by an increasing number of social scientists for making causal inferences.

  1. Results

3.3. Cross-lagged panel analyses

The results obtained are described in each of the figures, except in Figure 6. I consider it necessary to include a brief paragraph describing the results of Figure 6.

Response from Authors:

Thank you for catching the error. In the current version all the results presented on Figures 1-6 are described with proper references to the figures. Actually, all the descriptions were already there in the previous version but we unwittingly omitted the reference to fig 3, and, in consequence, all the consecutive Figure numbers were mismatched.

Discussion

The preparation of the discussion should be more adjusted to the formulated working hypotheses. It would be easier to trace the arguments adduced to the results obtained. indicate whether or not the hypothesis in question is accepted.

Response from Authors:

We have revised the discussion aiming to clarify whether or not the particular hypotheses were supported by the data. We hope that in the present form the discussion in clearer.

  1. Limitations

The participants' age ranged from 18 to 63. It is possible that the age of the participants and the sex may have influenced the results?

I believe that the authors should include some comment on this.

Response from Authors:

Thank you for this point. As each of the three Reviewers of the paper referred to this issue (i.e., the lack of adding age and gender as control variables in the tested models), we decided to re-run all the analyses with age and gender controlled for. Thus, in the current version we additionally provide partial correlations with age and gender controlled (see Table 1, values above the diagonal). Moreover, across all the cross-lagged analyses we now present the alternative set of coefficients, derived from the models run with age and gender controlled for (see fig 1-6; values presented in the brackets). Importantly, adding these two control variables did not change any of the relevant results. We have now commented on this issue in the manuscript.

References                                                                           

- Check the references. Some do not follow the format of the magazine. For example:

  1. Hansenne, M. Valuing Happiness is Not a Good Way of Pursuing Happiness, but Prioritizing Positivity is: A Replication Study. Psychol. Belg. 2021, 61(1), pp. 306–314. doi: https://doi.org/10.5334/pb.1036

Delete “pp”

Delete “doi”

  1. Cohen, J. Statistical Power Analysis. Curr. Dir. Psychol. Sci. 1992;1(3):98-101. doi:10.1111/1467-8721.ep10768783

Delete “ ; ”

Response from Authors:

Thank you for pointing that out; we have corrected these and other references to meet the IJEPRH requirements.

To sum up, after explaining the strengths and weaknesses in the text, I recommend publishing the article after the indicated corrections.

Reviewer 3 Report

Content: This is an interesting study based on data of 206 adult participants who completed six questionnaires twice (one-year interval) measuring time perspectives, gratitude, savoring the moment, prioritizing positivity, satisfaction with life, positive and negative affect. The authors describe in their results section how the various parameters influenced each other.

Comment: The study design and the description of the methods, the presentation of the results and the discussion are excellent from a scientific and theoretical point of view. It is fantastic to see the results for the various models. However, I have two major comments:

  • Did the authors consider age and sex as influencing variables? Age and sex are known. Thus, they should be statistically considered. What influence did age and sex have on the results?
  • In the conclusion section there should be at least one paragraph that explains much clearer to a reader who is not familiar with model calculations what the main two or three findings of these study are and how they can be applied in practice.

Author Response

Content: This is an interesting study based on data of 206 adult participants who completed six questionnaires twice (one-year interval) measuring time perspectives, gratitude, savoring the moment, prioritizing positivity, satisfaction with life, positive and negative affect. The authors describe in their results section how the various parameters influenced each other.

Comment: The study design and the description of the methods, the presentation of the results and the discussion are excellent from a scientific and theoretical point of view. It is fantastic to see the results for the various models. However, I have two major comments:

  • Did the authors consider age and sex as influencing variables? Age and sex are known. Thus, they should be statistically considered. What influence did age and sex have on the results?

Response from Authors:

Thank you for this point. As each of the three Reviewers of the paper referred to this issue (i.e., the lack of adding age and gender as control variables in the tested models), we decided to re-run all the analyses with age and gender controlled for. Thus, in the current version we additionally provide partial correlations with age and gender controlled (see Table 1, values above the diagonal). Moreover, across all the cross-lagged analyses we now present the alternative set of coefficients, derived from the models run with age and gender controlled for (see fig 1-6; values presented in the brackets). Importantly, adding these two control variables did not change any of the relevant results of our study. We have now commented on this issue in the manuscript.

  • In the conclusion section there should be at least one paragraph that explains much clearer to a reader who is not familiar with model calculations what the main two or three findings of these study are and how they can be applied in practice.

Response from Authors:

We have developed the closing part of the paper in order to respond to this point. The paragraph now reads as follows:

"The importance of well-being for public is increasingly accepted [75], but the relevance of individual differences in personality for public health policy has not received sufficient attention [76]. The present data reaffirm the importance of TPs as influences on well-being, consistent with Zimbardo and Boyd’s [8] account of the importance of adaptive TPs for societal flourishing. The data also show that TP may impact activities that support well-being, especially gratitude. In principle, the implementation of interventions that elevate gratitude and other WBBs may contribute to personal and societal well-being [77], consistent with positive psychology perspectives [18, 19]. However, over a one-year timespan, we could not substantiate a direct causal effect of WBBs on well-being, implying that more research is necessary to determine the circumstances under which practical interventions based on WBB activities are effective. In short, we need to know how long it takes for activities to impact well-being to adequately evaluate the impacts of interventions.

The current findings also suggest that interventions focused on WBBs are unlikely to directly change individuals’ TPs, which will continue to impact well-being post-intervention. However, the efficacy of interventions may be moderated by the person’s pre-existing TPs. Our previous report [6] demonstrated compensatory effects that imply interventions based on WBBs may be most effective for individuals with maladaptive TPs. Although TPs may be difficult to change, practitioners may be able to tailor interventions to individual vulnerabilities associated with personal time horizon”

Round 2

Reviewer 1 Report

The Authors have satisfactorily addressed all my concerns. I recommend the paper for publication.

Author Response

thanks very much